# Improved Media Formulations for Primary Cell Cultures Derived from a Colonial Urochordate

**DOI:** 10.3390/cells12131709

**Published:** 2023-06-23

**Authors:** Andy Qarri, Dietmar Kültz, Alison M. Gardell, Baruch Rinkevich, Yuval Rinkevich

**Affiliations:** 1Helmholtz Zentrum München, Regenerative Biology and Medicine Institute, 81379 Munich, Germany; 2Department of Animal Sciences, University of California Davis, Davis, CA 95616, USA; 3School of Interdisciplinary Arts and Sciences, University of Washington Tacoma, Tacoma, WA 98402, USA; 4Israel Oceanographic & Limnological Research, National Institute of Oceanography, Tel Shikmona, P.O. Box 9753, Haifa 3109701, Israel

**Keywords:** *Botryllus schlosseri*, primary cultures, blood cells, cell culture media, cell proliferation, cell viability

## Abstract

The cultivation of marine invertebrate cells in vitro has garnered significant attention due to the availability of diverse cell types and cellular potentialities in comparison to vertebrates and particularly in response to the demand for a multitude of applications. While cells in the colonial urochordate *Botryllus schlosseri* have a very high potential for omnipotent differentiation, no proliferating cell line has been established in *Botryllus*, with results indicating that cell divisions cease 24–72 h post initiation. This research assessed how various *Botryllus* blood cell types respond to in vitro conditions by utilizing five different refinements of cell culture media (TGM1–TGM5). During the initial week of culture, there was a noticeable medium-dependent increase in the proliferation and viability of distinct blood cell types. Within less than one month from initiation, we developed medium-specific primary cultures, a discovery that supports larger efforts to develop cell type-specific cultures. Specific cell types were easily distinguished and classified based on their natural fluorescence properties using confocal microscopy. These results are in agreement with recent advances in marine invertebrate cell cultures, demonstrating the significance of optimized nutrient media for cell culture development and for cell selection.

## 1. Introduction

Countless attempts have been made to develop cell cultures from marine invertebrates. Since the 1960s, over 500 peer-reviewed publications have been published on this topic, with the main focus being on six phyla, which are Porifera, Cnidaria, Crustacea, Mollusca, Echinodermata, and Urochordata [1,2,3,4,5,6,7]. However, until recently [8], not a single continuous cell line was established for any aquatic invertebrate taxon, as all efforts to obtain lasting proliferating cultures from marine invertebrates have inexplicably failed [1,2,9], despite the increasing demand of these cultures for a wide range of applications [5,6,7,8,10,11]. Further, it has been repeatedly shown that primary cultures of marine invertebrate cells cease dividing 24–72 h from onset [1,2]. It is worth noting that a substantial number of unsuccessful attempts to establish primary cell cultures from marine invertebrates are not reported in the scientific literature, resulting in incomplete or fragmented knowledge of effective in vitro protocols [7,12], and researchers often end up revisiting unsuccessful methods and futile experimental protocols [1,2]. Considering the above discussion, a recent advancement in sponge cell culture has emphasized the significance of developing optimized nutrient media for the development of primary cell cultures [4,5,6,7,8,9,10,11,12,13], a research approach that has led to the succeeded establishment of long-lasting cultures from several sponge species [8,13].

*Botryllus schlosseri* (Chordata, Tunicata, Ascidiacea), a colonial urochordate of cosmopolitan distribution, serves as a valuable model organism in various scientific disciplines, including stem cell biology [14,15], which has garnered significant attention due to the exceptional cellular potentialities exhibited by various cell types from this species, such as multipotency and totipotency [5,16,17,18,19,20,21,22], all of which hold great promise for in vitro applications. Colonies of *B. schlosseri* exhibit both tight asexual (referred to as blastogenesis) and sexual modes of reproduction [14,15]. After fertilization of the eggs, tadpole larvae are released and swim for approximately half an hour before attaching to substrates, metamorphosing and developing into the first colonial modules, known as oozooids, that bud the subsequent generations of zooids [23,24]. In each *B. schlosseri* colony, the zooids create typical star-shaped systems, all embedded within the colony’s tunic, a transparent gelatinous matrix of the colony that also holds the colonial circulatory system, connecting all zooids and terminating at the periphery of the colony with blind vasculature termini, called ampullae. *B. schlosseri* colonies develop through the asexual mode of reproduction by undergoing weekly blastogenic cycles, which involve series of growth and death of the zooids. Each weekly blastogenic cycle consists of four major stages (A to D; sensu [25]), during which sets of primary buds mature to adult zooids, while secondary buds emerge from the body walls of primary buds. Concurrently, all functional zooids undergo resorption through massive apoptotic and phagocytosis events of all functional zooids [26], with properties indicating a high capacity for stem cell activity throughout life [27,28].

Several studies have attempted to develop primary cultures from various cell types of *B. schlosseri*, including blood cells [10,29,30,31,32], cells originated from epithelial layers that exhibit de novo stemness signatures [18,30,33,34,35], and cells originating from embryos [17]. Some of these studies have also focused on optimizing the type of medium and media additives such as growth factors [29,36]. Despite these attempts, no proliferating primary cell cultures from *Botryllus* have been established, with no cell divisions observed beyond 72 h after culture initiation. In numerous cases, cultures were also overgrown by opportunistic microorganisms, including bacteria and protists [1,2,17,29,31,32,37]. In response to the above challenges, and in line with the approach taken by Munroe et al. [4] and Conkling et al. [13], the present study aimed to (a) enhance our understanding of the in vitro conditions that promote *B. schlosseri* cell growth, by utilizing five media formulations, and (b) provide an assessment of the proliferation status of cultured cells in the studied media.

## 2. Materials and Methods

### 2.1. B. schlosseri Husbandry

*B*. *schlosseri* colonies were collected from the rocky intertidal zone at Helgoland Island (Germany), tied on 5 × 7.5 cm glass slides and reared for 3 days at the Biological Station Helgoland of the Alfred Wegener Institute, transferred to the Institute of Regenerative Biology and Medicine, Munich, and kept vertically in slots of glass staining racks, in 5 L plastic tanks (22.5 °C) supplied with a standing artificial sea water system (ASW, [31,32]) and aerated by air stones. Colonies were fed daily with dried algae powder and commercial food (Reef energy plus, Red Sea, Ltd., London, UK). ASW was changed three times a week, and colonies were gently cleaned twice a week using small, soft brushes to remove trapped food particles, fouling organisms, and debris. All experimental colonies were in good health and well adapted to their maintenance conditions.

### 2.2. B. schlosseri Cell Extraction under Aseptic Conditions

Colonies were first taken out from aquaria and photographed under a stereomicroscope (Leica M50 equipped with a camera Leica DFC310 FX, Leica Microsystems, Wetzlar, Germany), and cells were extracted from *B*. *schlosseri* as described [32], under aseptic conditions using washing solution (WS), Merck Millipore, Darmstadt, Germany. WS [32] was made with autoclaved ASW, 0.2 µm filtered (Corning; Cat. No. 431218), supplemented with a penicillin streptomycin mixture (Gibco; 15070-063) or with penicillin, streptomycin, amphotericin b, 10,000 IU/mL, 10 mg/mL, 25 µg/mL (MP Biomedicals; Cat. No. 1674049), and gentamicin (50 mg/mL; Gibco; 15750-037) and stored at room temperature. Three WS variations were prepared containing ASW and antibiotic stock solutions: WS1 contained 88% ASW and 12% penicillin streptomycin; WS2 contained 88% ASW, 6% penicillin, streptomycin, amphotericin b, and 6% gentamicin; WS3 contained 88% ASW and 12% gentamicin.

### 2.3. Blood Cell Observations

Primary blood cell cultures were observed once every other day, where cells were counted using a hemocytometer and photographed under the microscope (Primo Vert, Zeiss inverted system microscope, equipped with a camera). Cell viability was determined using Trypan Blue Solution (Gibco; Cat. No. 15250-061, Thermo Fisher Scientific, Waltham, MA, USA). Contamination by thraustochytrids [1,31], single-celled saprotrophic eukaryotes, was monitored once every other day by carefully observing culture samples under the microscope, while the presence of yeast and fungi was detected by Calcofluor White Stain (Sigma-Aldrich; 18909, St. Lois, MI, USA), according to the manufacturer’s instructions. Samples were visualized under a fluorescent microscope with an emission range of 300 to 440 nm.

### 2.4. Media

*B*. *schlosseri* cultures were maintained in a liquid growth medium (tunicate growth medium; TGM) containing ASW, basal media (DMEM/F-12[HAM] 1:1, Dulbecco’s Modified Eagle Medium/Nutrient Mixture F-12, Gibco, Cat. No. 11320-074; RPMI medium 1640, Gibco, Cat. No. 21875-034), foetal bovine serum (FBS qualified HI, Brazil, Gibco, Cat. No. 10500-064), antibiotics, l-glutamine solution 200 mM (Sigma-Aldrich, 59202C), sodium pyruvate solution (Sigma-Aldrich, S8636), and HEPES buffer solution 1M (Sigma-Aldrich, H0887). Five medium variants (Appendix A) were then prepared: (1) medium TGM1 (pH = 7.21) contained 1% of l-glutamine solution 200 mM, 1% of HEPES buffer solution 1M, 1% of penicillin streptomycin, 1% of sodium pyruvate solution, 10% of foetal bovine serum, and 86% of DMEM/F-12[HAM] 1:1; (2) medium TGM2 (pH = 7.13) contained 1% of l-glutamine solution 200 mM, 1% of HEPES buffer solution 1M, 1% of penicillin streptomycin, 1% of sodium pyruvate solution, 10% of foetal bovine serum, and 86% of RPMI medium 1640; (3) medium TGM3 (pH = 6.76) contained 1% of l-glutamine solution 200 mM, 1% of HEPES buffer solution 1M, 1% of penicillin streptomycin amphotericin b, 1% of gentamicin, 1% of sodium pyruvate solution, 10% of foetal bovine serum, and 85% ASW; (4) medium TGM4 (pH = 7.01) contained 1% of l-glutamine solution 200 mM, 1% of HEPES buffer solution 1M, 1% of gentamicin, 1% of penicillin streptomycin, 1% of sodium pyruvate solution, 10% of foetal bovine serum, 63% of DMEM/F-12[HAM], and 22% of ASW; (5) medium TGM5 (pH = 7.14) contained 1% of l-glutamine solution 200 mM, 1% of HEPES buffer solution 1M, 1% of penicillin streptomycin, 1% of sodium pyruvate solution, 20% of foetal bovine serum, 38% of DMEM/F-12[HAM], and 38% of ASW. All media were stored at 4 °C and were used within five days.

### 2.5. General Culture Conditions

Media were filtered (0.2 µm) before use, all glassware was autoclaved, and only sterilized plasticware was used. Fifteen experiments on primary cultures (cells growing in suspension) were performed and kept for 2 to 4 weeks, with each containing lumped *B. schlosseri* blood cells from 2–4 colonies (total 40 colonies) in blastogenic stages A–D. There were no discernible differences between cell types extracted from various blastogenic states, as shown in a previous study [32]. Extracted cells were evenly distributed into multiple 35 mm Petri dishes (Thermo Fisher Scientific Nunc, 171099; 0.5 × 10^6^ cells/mL, 1.5 × 10^6^ cells/dish) containing 3 mL of liquid medium and were incubated (Innova 42) at 20 °C under normal atmosphere conditions. The medium was changed every other day, where the contents of the dishes were collected into a 15 mL tube, washed 3 times with fresh medium, followed by centrifugation (1000× *g*, 10 min), and seeded to new dishes with fresh medium. The criterion for experiment termination was ≤ 50% viability of total recovered cells.

### 2.6. Blood Cell Characterization

We used 12 colonies in blastogenic stages A–D (A = 3, B = 3, C = 3, and D = 3). Prior to cell extraction, each colony was photographed under a fluorescent microscope (Leica M205 FCA equipped with a camera Leica DFC9000 GT) using four channels (blue: Ex 325–378 nm, Em 438–485 nm; green: Ex 450–490 nm, Em 500–550 nm; red: Ex 540–580 nm, Em 593–667 nm; and far red: Ex 590–650 nm, Em 663–737 nm). Images were obtained using Leica software (LAS X version: 3.6.0.20104). Post cell extractions, the blood cells in ASW were imaged in 35 mm glass-bottom dishes (Ibidi, Cat. No. 81218) using a laser scanning confocal microscope (Zeiss LSM710) with four channels (blue: Ex 405 nm, Em 453 nm; green: Ex 488 nm, Em 536 nm; red: Ex 561 nm, Em 607 nm; and far red: Ex 633 nm, Em 697 nm), and images were obtained using Zen 2.3. The intensity of the fluorescent signal per cell was calculated using Fiji software (http://imagej.nih.gov/ij (accessed on 2 May 2023)). Each image was inverted to a grey scale followed by the measurement of the light signal emitted by cells in relative fluorescent units (RFU). For cell identification in cultures, primary cultures were sampled, and cells were stained with fluorescent dye Hoechst 33342 (Thermo Fisher, Cat. No. H3570). Then, cell pellets were washed with ASW (X3), followed by centrifugation at 1000× *g* for 10 min, and stained with DiD (Vybrant, V22887) for membrane labeling. Cell pellets were washed (three times) with ASW and centrifuged (1000× *g*, 10 min). The cells were then seeded on glass coverslips (coated with 0.01% Poly L Lysine solution, Sigma-Aldrich P4707, according to the manufacturer’s instructions), and inserted at the bottoms of 12 well plates, left to adhere for approximately 4 h. Then, each well containing stained cells was fixed for 30 min with 4% paraformaldehyde (PFA; Thermo Fisher Scientific, Cat. No. 043368.9M) at room temperature, coverslips were mounted on cover glass slides, and cells were observed and photographed under an epifluorescence microscope (Zeiss AxioImager2, Zeiss, Oberkochen, Germany). Images were obtained using Zen 2.3.

### 2.7. Hematoxylin and Eosin (H&E) Staining

The H&E staining protocol was performed on blood cell samples that were seeded on glass coverslips (coated with 0.01% Poly L Lysine solution) at the bottoms of 12 well plates and left to adhere. Then, each well was fixed with 4% PFA. Mayer’s Hematoxylin solution (Sigma; Cat. No. MHS32) and 0.5% Eosin Y-solution (Sigma, Cat. No. 1.09844.1000) were employed according to the manufacturer’s instructions. Coverslips were mounted on cover glass slides, and cells were observed and photographed under an epifluorescence microscope (Zeiss AxioImager2) with bright field. Images were obtained using Zen 2.3.

### 2.8. Immunofluorescence Staining

Blood cell samples were seeded on glass coverslips that were inserted at the bottom of 12 well plates and coated with 0.01% Poly L Lysine solution and left to adhere. Each well was then fixed with 4% PFA. Then, samples were washed two times with PBS, were permeabilized in 0.1% Triton x-100 in PBS for 10 min at 4 °C, and were re-washed two times with 0.02% Tween-20 in PBS. Nonspecific binding sites were blocked by incubation in 5% bovine serum albumin diluted in 0.02% Tween-20 in PBS for 60 min at 4 °C. The samples were incubated overnight at 4 °C with primary anti PCNA (Proliferating Cell Nuclear Antigen) antibody developed against rabbit (Abcam, ab18197, 1:1000), washed with 0.02% Tween-20 in PBS (2 × 10 min), and incubated with secondary antibodies (Invitrogen, Alexa fluor 568 donkey anti rabbit, Cat. No. 10042, 1:1000) for 120 min at 4 °C. The samples were then washed twice with PBS, and the coverslips were mounted on slides using Fluoromount-G with DAPI (Invitrogen, Cat. No. 00-4959-52, Waltham, MA, USA). Negative controls for each experiment were established on coverslips that were incubated with blocking solution, lacking primary antibodies, and were exposed only to secondary antibodies. Cover slides were photographed under an epifluorescence microscope (Zeiss AxioImager2). Images were obtained using Zen 2.3. The counting of total stained cells was conducted using Fiji software (http://imagej.nih.gov/ij (accessed on 2 May 2023)).

### 2.9. Statistics

Statistical analyses were conducted on the cell types using SPSS V16. A one-way ANOVA test using a post hoc comparison (Bonferroni and Tukey HSD) was applied on the cell types, including PCNA^+^ cells, at four time points (day 0, 24 h, days 3, 8). 

## 3. Results

### 3.1. Identification of B. schlosseri Blood Cell Populations

Twelve *B. schlosseri* colonies (blastogenic stages: A = 3, B = 3, C = 3, D = 3) were used. Prior to cell extractions, the natural auto-fluorescence of the blood cell emissions was evaluated by capturing images of each colony using blue, green, red, and far-red channels under a fluorescent microscope. The *B. schlosseri* tunic matrix did not autofluoresce. However, distinct blood cell populations emitted varying levels of natural fluorescence (Figure 1). This prompted us to use confocal microscopy to examine in detail the extracted blood cells (Figure 2) and to calculate fluorescence intensity. The results showed that haemoblasts (*n* = 12 cells; Figure 2(a1–5)) exhibited fluorescence intensity values of 0, 32.6 ± 16.4, 59.7 ± 14.07, and 141.16 ± 32.3 [RFU] for blue, green, red, and far-red channels, respectively. Morula cells (*n* = 28; Figure 2(b1–5)) showed fluorescence intensity values of 2.1 ± 1.02, 17.01 ± 20.7, 46.7 ± 19.8, and 61.3 ± 34.8 RFU for blue, green, red, and far-red channels, respectively. Pigment cells (*n* = 15; Figure 2(c1–5)) showed fluorescence intensity values of 4.3 ± 1.3, 22.4 ± 10.2, 19.3 ± 4.6, and 40.92 ± 12.01 RFU for blue, green, red, and far-red channels, respectively. Nephrocytes (*n* = 17; Figure 2(d1–5)) exhibited fluorescence intensity values of 0.75 ± 0.9, 23.5 ± 9.95, 12.9 ± 4.3, and 45.3 ± 14.13 RFU for blue, green, red, and far-red channels respectively. In addition, we observed the following two life stages of thraustochytrid cells that were present in the cultures: mononucleated cells (*n* = 4; Figure 2(e1–5)) that showed fluorescence intensity values of 4.46 ± 5.1, 4.11 ± 6.45, 22.28 ± 7.97, and 2.93 ± 3.51 RFU for blue, green, red, and far-red channels, respectively, and sporangia cells (*n* = 9; Figure 2(f1–5)) that showed fluorescence intensity values of 4.6 ± 3.85, 29.5 ± 14.35, 18.25 ± 5.04, and 9.3 ± 10.54 RFU for blue, green, red, and far-red channels, respectively.

Isolated haemocytes of *B. schlosseri* (from blastogenic A–D colonies) and thraustochytrids were subsequently identified using histological (H&E), nuclear (Hoechst), and membrane (DiD) staining (Figure 3, Figure 4 and Figure 5). The haemoblasts (4–6 µm; Figure 3a,b) are small, brown, spherical blood cells. They contain a circular nucleus positioned in the middle of the cell and a nucleolus (often non-recognizable) stained (Figure 4a) in blue while surrounded by a thin layer of basophilic cytoplasm (Figure 5a–d). The macrophage-like cells (10–20 µm) are large ameboid phagocytes, marked (using H&E) as orange, red, pink, and blue colored cells, with an elliptical nucleus at the periphery of the cell and recognizable brown, yellow, and black vacuoles (Figure 3c,d) that occupy most of the cells’ volumes (Figure 4b and Figure 5e–h). Morula cells (7–11 µm) are spherical in shape, with a barely recognizable small nucleus (2 µm) situated at the periphery of the cell, possessing diverse brown, yellow, and black vacuoles (Figure 3e,f). H&E stained these cells in orange, pink, and blue colors, with recognizable vacuoles that occupy most of the cell volume (Figure 4c and Figure 5i–l). Granular amoebocyte cells (7–15 µm) are oval shaped, containing a small nucleus (2 µm) and micro/macro granules, with vacuoles stained with orange, pink, red, and blue colors (Figure 4d and Figure 5m–p). The pigment cells (7–20 µm) are spherical and granular in appearance and contain brown granules of varying sizes, with colors diverged from dark blue or brown to black that exhibit Brownian motion (Figure 3g,h). These cells were stained (H&E) as pink, red, and blue colors, and the round nucleus appeared in different positions within the cell (Figure 4e and Figure 5q–t). The nephrocytes (7–20 µm) share similar staining characteristics as pigment cells. These cells also contain brown and yellow granules (Figure 3i,j), exhibiting (H&E) pink and blue colors (Figure 4f and Figure 5u–x).

We further observed two distinct life stages of thraustochytrid cells, which may appear as cell clusters after nine days in vitro. The mononucleated cells (4–10 µm) are spherical and smooth, containing a small, hardly recognizable nucleus (1–2 µm). These cells appeared in clusters (up to 200 µm) composed of at least 3 cells, each. H&E staining marked the cell cytoplasm as pink, and the membranes surrounding the cell were stained with a thin blue line (Figure 4g). The sporangia cells (10–200 µm) are dark brown, spherical cells with a grainy texture and a barely recognizable nucleus. Stained (H&E) cells appeared pink with patterns of blue (Figure 4i).

### 3.2. Cultivation of B. schlosseri Blood Cells

We used 40 *B. schlosseri* colonies (7–15 zooids per colony) in blastogenic stages A–D (Table 1). Blood cells were extracted from the marginal ampullae and washed with one of three types of washing solutions (WS1, WS2, WS3; Table 1), were subjected to specific antibiotic combinations, and were then cultured in suspensions of one of five medium variants (TGM1- TGM5). In total, 15 experiments were carried out with high cell viability observed at onset (87.08–94.65%; Table 1). Cultures were observed every second day for up to 26 days (Figure 6), and four time points (0, 24 h, days 3, 8) were specifically studied by counting cell types (Figure 7) and assessing the proliferative status using PCNA (Figure 8 and Appendix A).

#### 3.2.1. Primary Cultures: TGM1 Medium

Cell cultures were studied for 18 days (Figure 6a) with minimal contamination events (just two [fungi] out of 17 plates, on days 12 and 14; Appendix A). At the beginning of the study (day 0; Appendix A), one-way ANOVA revealed significant differences (*p* < 0.001) in the distributions of cell types, which formed three distinct groups (morula; macrophage-like and pigment cells; granular amoebocyte cells; Figure 7a) and four distinct groups of PCNA^+^ cell dispersals (haemoblasts and macrophage-like cells; morula and pigment cells; nephrocyte cells; granular amoebocyte cells; Figure 8a and Appendix A). The same three groups of cell types (*p* < 0.001; Figure 7f) were recorded at the 24 h, 3 day, and 8 day time points (Figure 7k,p and Appendix A), while non-significant differences (*p* > 0.05) were recorded in PCNA^+^ cell distributions at the 24 h time point (Appendix A), PCNA^+^ haemoblasts and macrophage-like cells (36.8 ± 0.58% and 28.9 ± 3.3%, respectively; Figure 8f) outnumbered other cell types. At day three (Appendix A), we recorded different (*p* < 0.001) PCNA^+^ cell distributions (haemoblasts and macrophage-like cells; nephrocyte cells; pigment cells; granular amoebocyte cells; Figure 8k and Appendix A). At day 8 (Appendix A), only two distinct groups of PCNA^+^ cells were recorded (*p* < 0.05; haemoblasts and nephrocyte cells, respectively; Figure 8p and Appendix A). Cell viability was reduced from >90% to 84% at days 4–6 (Figure 6f), with cell populations dominated by macrophage-like cells, morula cells, and pigment cells. From day 10 to 18, cell numbers further decreased from 1.99 × 10^6^ ± 0.14 to 0.83 × 10^6^ ± 0.16 cells/mL^−1^, respectively, and a sharp decline of viability was observed from 78.17% ± 1.5 to 48.4% ± 0.8, respectively (Figure 6a,f and Appendix A).

#### 3.2.2. Primary Cultures: TGM2 Medium

Cell cultures were studied for 16 days (Figure 6b), with minimal contamination events (three [×2 thraustochytrids, yeasts] out of nine plates, on days 10 and 14; Appendix A). No significant difference (*p* > 0.05; one-way ANOVA) was recorded at the beginning (day 0; Appendix A) in the distributions of cell types with macrophage-like cells and morula cells (24.4 ± 5.05% and 37.3 ± 5.3%, respectively), comprising >61% of total cell numbers (Figure 7b) and macrophage-like cells portrayed 52.56 ± 11.9% of all PCNA^+^ cells (*p* > 0.05, one-way ANOVA; Figure 8b and Appendix A). At the 24 h time point, morula cells and macrophage-like cells continued as the most abundant (>64%) cell types (24.6 ± 3% and 38.6 ± 8%, respectively; Figure 7g; *p* > 0.05, Appendix A), while macrophage-like cells were the most abundant cell type, with a reduction from onset (43.2 ± 10.9%) of all PCNA^+^ cells (*p* > 0.05; Figure 8g and Appendix A). At day three, macrophage-like cells, morula cells, and pigment cells (31.15 ± 0.95%, 20.23 ± 0.3%, and 31.62 ± 0.28%, respectively; *p* > 0.05, Appendix A) comprised >83% of total cell numbers (Figure 7l), while macrophage-like cells, morula cells, and pigment cells (18.9 ± 9.7%, 12.4 ± 3.9%, and 49.9 ± 6.6%, respectively) comprised >81% of all PCNA^+^ cells (*p* > 0.05; Figure 8l and Appendix A). At day 8 (*p* > 0.05; one way ANOVA; Appendix A), macrophage-like cells, morula cells, and pigment cells (22.19 ± 0.27%, 23.94 ± 0.1%, and 35.66 ± 2.3%, respectively) comprised >81% of total cell numbers (Figure 7q), and macrophage-like cells, morula cells, and pigment cells (15.48 ± 1.7%, 60.71 ± 15.2%, and 23.8 ± 13.47%, respectively) comprised >99% of all PCNA^+^ cells (*p* > 0.05, one-way ANOVA; Figure 8q and Appendix A). Cell viability values were reduced from >83% to 77% at days 4–6 (Figure 6g), with cell populations dominated by haemoblasts, macrophage-like cells, morula cells, and pigment cells. From days 10 to 16, cell numbers further decreased from 1.29 × 10^6^ ± 0.13 to 0.63 × 10^6^ ± 0.15 cells/mL^−1^, and a sharp decline of viability was observed, from 68.15% ± 1.1 to 50.2% ± 0.7 (Figure 6b,g and Appendix A).

#### 3.2.3. Primary Cultures: TGM3 Medium

Cell cultures were studied for 12 days (Figure 6c) with no contamination. At day 0 (Appendix A), significant differences (*p* < 0.001; one-way ANOVA) were recorded in cell type distributions that formed three distinct groups (macrophage-like cells and morula cells; haemoblasts and pigment cells; nephrocytes and granular amoebocyte cells; Figure 7c), while non-significant differences (*p* > 0.05) obtained for PCNA^+^ cells with haemoblasts and macrophage-like cells (31.06 ± 7.7% and 41.11 ± 4.02%, respectively; Figure 8c and Appendix A). At the 24 h time point (Appendix A), cell type distributions formed four distinct groups (*p* < 0.001; morula cells; macrophage-like and pigment cells; haemoblasts; granular amoebocyte and nephrocyte cells; Figure 7h) and two distinct groups of PCNA^+^ cells (haemoblasts and macrophage-like cells; granular amoebocyte, morula, pigment, and nephrocyte cells; Figure 8h and Appendix A). At day three (Appendix A), we recorded four distinct cell groups (*p* < 0.001; morula and pigment cells; macrophage-like cells; haemoblasts; granular amoebocyte and nephrocyte cells; Figure 7m) and two distinct PCNA^+^ cell groups (haemoblasts; granular amoebocyte cells, morula cells, and nephrocyte cells; Figure 8m and Appendix A). At day 8 (Appendix A), we recorded three cell type groups (morula and pigment cells; macrophage-like cells; haemoblasts and nephrocyte cells; *p* < 0.001; Figure 7r) and two distinct PCNA^+^ groups (pigment cells; morula and nephrocyte cells; *p* < 0.05; Figure 8r and Appendix A). Along this period, the only observed change in cellular morphologies was of storage cells (pigment and nephrocyte cells) that transformed from oval to elongate structures, which is consistent with the results of Rinkevich and Rabinowitz [29]. Cell viability was reduced from >84% to 77% at days 4–6 (Figure 6h), with cell populations dominated by haemoblasts, macrophage-like cells, morula cells, and pigment cells. From day 10 to 12, cell numbers further decreased from 0.6 × 10^6^ ± 0.1 to 0.53 × 10^6^ ± 0.1 cells/mL^−1^, and a sharp decline of viability was observed from 55.3% ± 3.4 to 46.8% ± 2.7 (Figure 6c,h and Appendix A).

#### 3.2.4. Primary Cultures: TGM4 Medium

Cell cultures were studied for 26 days (Figure 6d) with no contamination. At day 0 (Appendix A), we recorded three distinct cell groups (*p* < 0.001; macrophage-like cells and morula cells; haemoblasts and pigment cells; granular amoebocyte and nephrocyte cells; Figure 7d) and two PCNA^+^ cell groups (*p* < 0.001; haemoblasts, macrophage-like cells, and pigment cells; granular amoebocyte, morula, and nephrocyte cells; Figure 8d and Appendix A). At the 24 h time point (Appendix A), no difference (*p* > 0.05) was recorded in cell type distributions, with haemoblasts, macrophage-like cells, and morula cells (17.58 ± 4.5%, 23.9 ± 0.5%, and 32.6 ± 2.5% respectively) comprising >74% of total cell numbers (Figure 7i), and four significant (*p* < 0.001) PCNA^+^ cell groups (haemoblasts; macrophage-like cells and pigment cells; granular amoebocyte, morula, and nephrocyte cells; Figure 8i and Appendix A) were recorded. Day three results (Appendix A) revealed a significant difference (*p* < 0.001) in cell type distributions compared to the onset and in PCNA^+^ cells when compared to 24 h. Four distinct cell groups (morula cells; macrophage-like cells and pigment cells; haemoblasts; granular amoebocyte and nephrocyte cells; Figure 7n) and two distinct PCNA^+^ groups (haemoblasts, macrophage-like cells, and pigment cells; granular amoebocyte, morula, and nephrocyte cells; Figure 8n and Appendix A) were present. At day 8 (Appendix A), five distinct (*p* < 0.001) cell type distributions were recorded (morula cells; pigment cells; macrophage-like cells; haemoblasts; granular amoebocyte and nephrocyte cells; Figure 7s), while there was no significant difference in PCNA^+^ cell distributions. Haemoblasts, macrophage-like cells, and pigment cells (41.3 ± 4.7%, 28.5 ± 8.1%, and 27.2 ± 5.6%, respectively) comprised 97% of all cell types (Figure 8s and Appendix A). Cell viability remained stable (>90% to 89%) at days 4–6 (Figure 6i), with cell populations dominated by macrophage-like cells, morula cells, and pigment cells. From day 10 to 14, cell numbers and viability remained stable with values of 2.4 × 10^6^ ± 0.42, 2.3 × 10^6^ ± 0.4 cells/mL^−1^, 84.9% ± 1.4, and 83.4% ± 1.3 (Figure 6d,i), and cell populations were dominated by haemoblasts, macrophage-like cells, morula cells, and pigment cells. From day 16 to 21, cell numbers decreased from 2.11 × 10^6^ ± 0.4 to 1.65 × 10^6^ ± 0.5 cells/mL^−1^, accompanied by a sharp decrease of viability from 79.98% ± 2.3 to 69.4% ± 2.6 (Figure 6d,i). Cell populations were dominated by haemoblasts and pigment cells (Appendix A). From day 22 to 26 (the fourth week), cell numbers further declined from 1.33 × 10^6^ ± 0.4 to 0.87 × 10^6^ ± 0.2 cells/mL^−1^, accompanied by a sharp decrease of viability from 62.6% ± 1.8 to 50. 5% ± 0.8 (Figure 6d,i). Cell populations were dominated mostly by pigment cells (Appendix A). 

#### 3.2.5. Primary Cultures: TGM5 Medium

Cell cultures were studied for 24 days (Figure 6e) with no contamination. At onset (day 0; Appendix A), five distinct cell type groups (*p* < 0.001) were recorded (morula cells; macrophage-like cells, and pigment cells; haemoblasts; nephrocyte cells; granular amoebocyte cells; Figure 7e), as well as two distinct (*p* < 0.001) PCNA^+^ cell groups (haemoblasts and macrophage-like cells; granular amoebocyte and nephrocyte cells; Figure 8e and Appendix A). After 24 h (Appendix A), no distinct cell type distributions (*p* > 0.05) were recorded, with haemoblasts, macrophage-like cells, morula cells, and pigment cells (20.5 ± 2.8%, 25.1 ± 3.5%, 23.5 ± 3.1%, and 20.5 ± 3.3%, respectively) comprising >89% of the total cell types (Figure 7j), while three distinct groups (*p* < 0.001; haemoblasts and macrophage-like cells; morula cells; granular amoebocyte, pigment and nephrocyte cells; Figure 8j and Appendix A) were recorded in PCNA^+^ cell distributions. At day three (Appendix A), we recorded two distinct groups in cell distributions (haemoblasts, macrophage-like cells, and morula cells; granular amoebocyte and nephrocyte cells; *p* < 0.05; Figure 7o) and three distinct groups (haemoblasts and macrophage-like cells; granular amoebocyte cells; pigment and nephrocyte cells; *p* < 0.001; Figure 8o and Appendix A) in PCNA^+^ cell distributions. At day 8 (Appendix A), two significant groups (haemoblasts, macrophage-like cells, morula cells, and pigment cells; granular amoebocyte and nephrocyte cells; *p* < 0.001; Figure 7t) were found in cell distributions, and no significant difference (*p* > 0.05) in the distributions of PCNA^+^ cells was observed, with haemoblasts, macrophage-like cells, and morula cells (46.9 ± 6.4%, 27.5 ± 9.9%, and 15.1 ± 3%, respectively) covering >89% of the total cells (Figure 8t and Appendix A). Cell viability values were stable (>93% to 92%) at days 4–6 (Figure 6j), with cell populations dominated by haemoblasts, macrophage-like cells, morula cells, and pigment cells. Cell numbers slightly decreased from day 10 to 14 (1.83 × 10^6^ ± 0.2 to 1.5 × 10^6^ ± 0.2 cells/mL^−1^), in concordance with a decrease in viability from 91.3% ± 2.5 to 86.9% ± 3.2 (Figure 6e,j), where cell populations were dominated by haemoblasts and pigment cells. From day 16 to 21, cell numbers further decreased from 1.24 × 10^6^ ± 0.1 to 0.94 × 10^6^ ± 0.2 cells/mL^−1^, and a sharp decrease of viability was observed from 83.8% ± 2 to 60.7% ± 1.8 (Figure 6e,j), dominated by haemoblasts and pigment cells (Appendix A). From day 22 to 24, cell numbers remained stable (0.8−0.9 × 10^6^ cells/mL^−1^), with a decrease in cell viability from 53.4% to 49.6% (Figure 6e,j), dominated mostly by pigment cells (Appendix A).

#### 3.2.6. Cell Types Changes in Primary Cultures—An Overview

The distribution of cell types varied among the five media and changed over time (Figure 7, Table 2). Initially (day 0), cell type distribution profiles exhibited a high degree of similarity, with macrophage-like cells and morula cells being the predominant cell types in all media. After 24 h, macrophage-like cells and morula cells remained the most prevalent cell types, but the presence of pigment cells (TGM3, TGM5) and haemoblasts (TGM5) was also observed. By day three, alterations in the distribution of cell types were noticed, with morula and pigment cells being more prominent in the TGM1 medium and macrophage-like cells and pigment cells being more prevalent in the TGM2 medium. However, the abundance profiles of TGM3, TGM4, and TGM5 media remained similar to that of the 24 h time point. By day 8, no further changes in cell type distribution were detected, and the abundance profiles of all five media were almost identical to those of day three.

#### 3.2.7. Proliferation of Primary Cultures—An Overview

PCNA^+^ activity varied among the five media and changed over time (Figure 8, Table 2). Initially, haemoblasts and macrophage-like cells were the most actively proliferating cell types in all media, with an additional case of PCNA^+^ activity in pigment cells in the TGM4 medium. After 24 h, we observed PCNA^+^ cell activity in haemoblasts, macrophage-like cells, and morula cells in TGM1 medium; macrophage-like cells in TGM2; haemoblasts and macrophage-like cells in TGM3; haemoblasts, macrophage-like cells, and pigment cells in TGM4; and haemoblasts and macrophage-like cells in TGM5. On day three, the most actively proliferating cell types varied by medium: haemoblasts and macrophage-like cells in TGM1; pigment cells in TGM2; haemoblasts in TGM3; haemoblasts and pigment cells inTGM4; and haemoblasts and macrophage-like cells in TGM5. By day 8, haemoblasts were the most actively proliferating cell types in TGM1, TGM4, and TGM5 media, while morula cells and pigment cells were the most actively proliferating in TGM2 and TGM3, respectively. 

## 4. Discussion

The world’s oceans and seas host hundreds of thousands of animal species, primarily invertebrates, with many types of cells that exhibit a wide range of cellular potentialities [38] and offer endless applications. Yet, while numerous cell lines have been commonly derived from vertebrates and terrestrial invertebrate taxa, e.g., insects and arachnids, until just recently [8], all efforts to develop cell lines from marine invertebrates have failed, despite many attempts made on a wide range of species over the past decades [1,2,3,6,7,9]. However, to establish a sustainable pharmaceutical industry based on the “blue economy”, the most viable approach is (a) to cultivate cells under controlled conditions by creating widespread “cell factories”, still an unaccomplished objective, and (b) to scale up cell cultures from marine organisms in bioreactors. These evolving stages not only allow the production of large supplies for any needed bioactive material, but they further make available the manufacturing of a diverse range of novel industrial bioproducts. The recent advances in marine invertebrate cell culture methodology supported by the development of optimized nutrient media for primary sponge cell cultures [4,13] have led the authors to establish long-lasting cultures from several species of sponges [8,13], revealing the importance of fine-tuning the medium for successful outcomes.

Following the importance of *B. schlosseri* as a model system in a wide range of biological disciplines [14,15,39,40,41] and the need to develop approved in vitro methodologies for its research (general approach in Rinkevich and Rabinowitz [36]), several research endeavors have centered around obtaining high cell yields; identifying appropriate factors for proper cell adhesion and attachment [30,32]; initiating primary cell cultures from embryos, epithelial cells, and circulating blood cells [10,17,18,29]; revealing impacts of media additives on primary cultures, such as growth factors [29]; establishing a defined medium for circulatory blood cells [29]; and evaluating the in vitro delayed stemness of extirpated colonial organs, including the emerged stemness signatures in epithelial monolayers [30,34,35]. All attempts to establish a proliferating cell line from *Botryllus* cells have been unsuccessful, with cells dividing for a brief period of 24–72 h post extraction [1,2,17,29]. To address these difficulties, the present study aimed to investigate the response of various types of *B. schlosseri* blood cells (haemoblasts, macrophage-like cells, granular amoebocytes, morula cells, pigment cells, and nephrocytes; [42,43]) under in vitro conditions. Specifically, we assessed cell type-specific responses using five versions of a basic medium during the initial few crucial weeks after initiation. To aid in cell identification, we utilized confocal microscopy and the differential autofluorescence of various cells, which provided new insights. For example, our findings indicated that haemoblasts were the only *B. schlosseri* cell type to exhibit a high signal in the far-red channel, which distinguishes them from thraustochytrid cells, and they lacked blue channel fluorescence, supporting Rinkevich and Rabinowitz’s [29] previous findings.

The different media variations (TGM1-TGM5) had distinct impacts on primary cultures of *B. schlosseri* blood cell types, reflected as changes in cell proliferation, viability, and dominant cell types. With regards to cell proliferation, we showed that the medium may stimulate the proliferation of distinct circulating cell types at different time points (onset, 24 h, 3 days, and 8 days). During this period, abundant cell types (haemoblasts, macrophage-like cells, morula cells, and pigment cells) exhibited varying activity patterns in different media that were formulated with varying proportions of basal media and ASW. For instance, TGM1 and TGM2 contained DMEM F12/HAM and RPMI, respectively, without ASW, whereas TGM3 contained ASW without DMEM F12/HAM or RPMI. In this study, cell proliferation was observed for at least 5 days, exceeding the documented 3 days post isolation in vitro barrier [2,29,32,37].

Similar to cell proliferation, differences in the viability of the cells were observed among the five media versions, resulting in variations in culture longevities. When setting 50% survivorship as a cut-off value, the TGM1, TGM2, and TGM3 cultures lasted for 18, 16, and 12 days, respectively, while TGM4 and TGM5 allowed for longer cultivation periods of 26 and 24 days, respectively. These findings align with those of [44], who conducted a short-term in vitro study on coral cells and found that cell viability decreased from 70% to 30% within the first week, as well as with Rinkevich and Rabinowitz [29], who studied *Botryllus* blood cell cultivation and observed a decrease in viability within four weeks. Rabinowitz and Rinkevich [30] reported shorter viability (7–9, 5–13, and 6–8 days) for *Botryllus* epithelial monolayers cultured with DMEM, RPMI, and HAM F12 on coated collagen 1 substrate. Regarding cellular morphology, this culturing criterion remained largely unchanged, except for the storage cells (pigment and nephrocyte cells) cultured in TGM3 medium (containing 85% ASW), which transformed from oval to elongated structures. This observation aligns with Rinkevich and Rabinowitz’s [29] findings that pigment cells undergo changes in shape when exposed to high salt concentrations in the culture medium.

The distribution of cell types varied among the five media and underwent alterations over time, similar to cell proliferation and viability. In the first 24 h, macrophage-like cells and morula cells were the most abundant cell types in all tested media. By day 3, while the abundance profiles of TGM3 and TGM4 media remained similar to that of 24 h, morula and pigment cells were more prominent in TGM1 medium. At day 8, no further changes in cell type distribution were observed, and the abundance profiles of all five media were almost identical to those of day three. As mentioned above, to the best of our knowledge, no study has yet revealed the distribution of *Botryllus* blood cell types under prolonged in vitro conditions. Our results show that, at onset, macrophage-like cells, granular amoebocyte cells, and morula cells varied between 23 and 28% each, while granular amoebocyte and morula cells varied between 40 and42% each. These results are in line with the literature [29,43] regarding the distributions of granular amoebocyte and morula cells but not for macrophage-like cells.

The findings of this study demonstrate that altering the basic culture medium can cause varying growth and proliferation rates among different types of cells, as observed in our study of *Botryllus* blood cells. These findings align with a recent study on the cultivation of sponge cells [4] that followed up with studies documenting the enhancement of the quantity and viability of sponge cells [8,13]. The results of the present study thus demonstrate that it is possible to culture *Botryllus* blood cells in vitro for up to one month in a consistent and reliable manner. During the initial week of culture, there was a noticeable medium-dependent increase in the proliferation of distinct blood cell types (could be further supported by increased mortality in other cell types), which eventually led, within less than one month from initiation, to the development of medium-specific primary cultures. This discovery may pave the way for the creation of various cell cultures, each consisting of distinct cell types. Further, the aforementioned outcomes were reinforced by the ease with which cell types could be identified and classified based on their natural fluorescence patterns using confocal microscopy, an additional tool, for improved cell type identification in the development of cell cultures from *B. schlosseri* circulating blood cells. Compared to mammalian and insect cultures, this approach is novel to marine invertebrate cell cultures.

## Figures and Tables

**Figure 1 cells-12-01709-f001:**
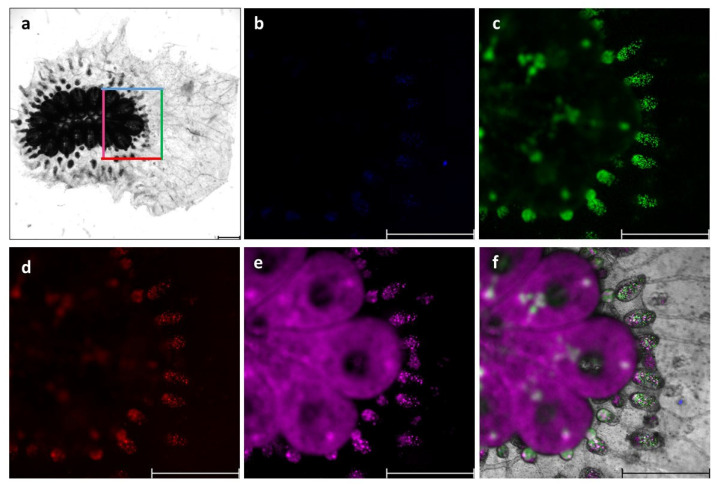
*B. schlosseri* whole-colony autofluorescence (**a**–**f**) performed on a single system with 15 zooids from a colony at blastogenic stage A: (**a**) Bright field image: the colored rectangle indicates the same highly magnified colonial area imaged under four channels (**b**–**e**); (**b**) the blue channel; (**c**) the green channel; (**d**) the red channel; (**e**) the far-red channel; (**f**) a merged high magnification area of a bright field image with the four channels. Scale bars: 1 mm.

**Figure 2 cells-12-01709-f002:**
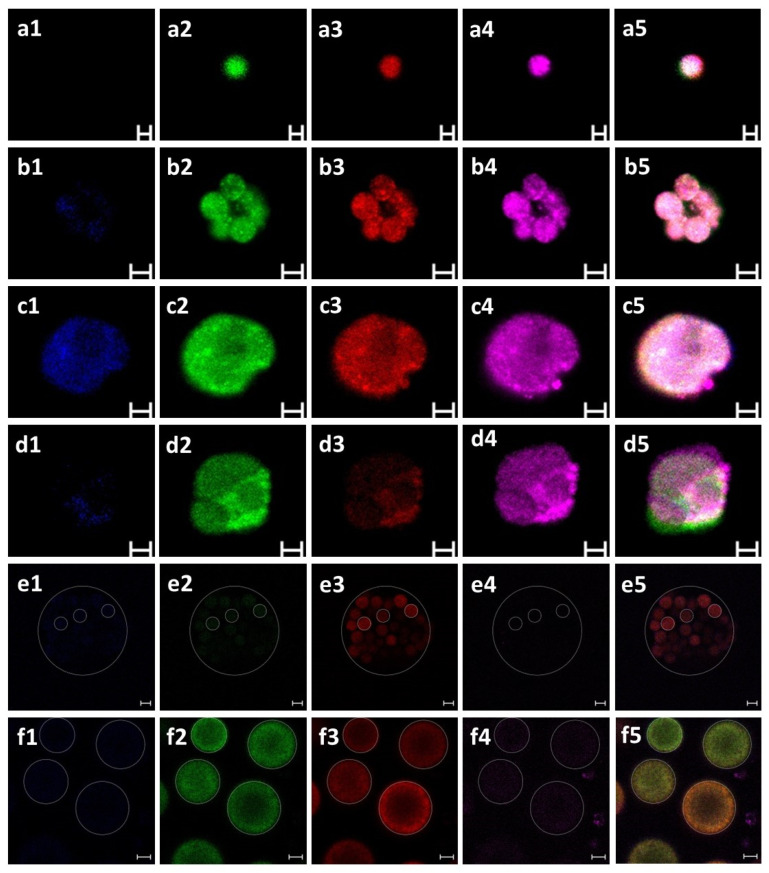
Confocal autofluorescence images in *B. schlosseri* and thraustochytrid cells at various wavelengths. (**a1**–**a5**) A hemoblast, an undifferentiated cell: blue (**a1**), green (**a2**), red (**a3**), and far-red (**a4**). (**a5**) Merged image of four channels. (**b1**–**b5**) A morula cell (immunocyte cell, belongs to the cytotoxic lineage): blue (**b1**), green (**b2**), red (**b3**), and far-red (**b4**). (**b5**) Merged image of four channels. (**c**–**d**) Storage cells. (**c1**–**c5**) Pigment cell: blue (**c1**), green (**c2**), red (**c3**), and far-red (**c4**). (**c5**) Merged image of four channels. (**d1**–**d5**) A nephrocyte cell: blue (**d1**), green (**d2**), red (**d3**), and far-red (**d4**). (**d5**) Merged image of four channels. (**e**,**f**) Thraustochytrid cells. (**e1**–**e5**) Clump of single multinucleated cells (dashed circle): blue (**e1**), green (**e2**), red (**e3**), and far-red (**e4**). (**e5**) Merged image of four channels. (**f1**–**f5**) Sporangia cells (dashed circles): blue (**f1**), green (**f2**), red (**f3**), and far-red (**f4**). (**f5**) Merged image of four channels. Scale bars: 2 µm in a, 5 µm in (**b**–**d**), 20 µm in (**e**,**f**).

**Figure 3 cells-12-01709-f003:**
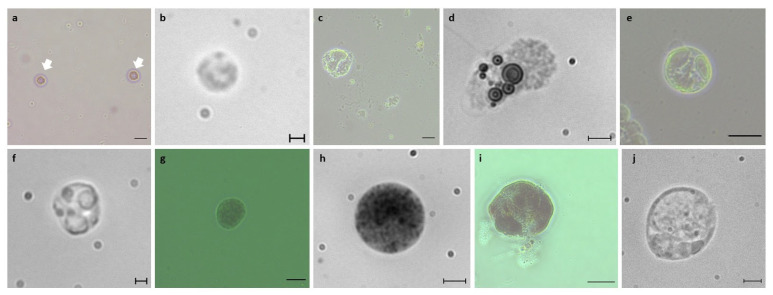
*B. schlosseri* live circulating blood cells under the microscope. (**a**,**b**) Haemoblasts: (**a**) two haemoblasts (arrows) under a bright field (**b**). (**c**,**d**) Macrophage-like cells: macrophage-like cell (**c**) under bright field (**d**). (**e**,**f**) Morula cells: morula cell (**e**) under a bright field (**f**). (**g**,**h**) Pigment cell: pigment cell imaged under the microscope (**g**) with a bright field (**h**). (**i**,**j**) Nephrocyte cell: nephrocyte cell imaged under the microscope (**i**) with a bright field (**j**). Scale bars: 2 µm in (**b**,**f**), 5 µm in (**d**,**h**,**j**), 10 µm in (**a**,**c**,**e**,**g**,**i**).

**Figure 4 cells-12-01709-f004:**
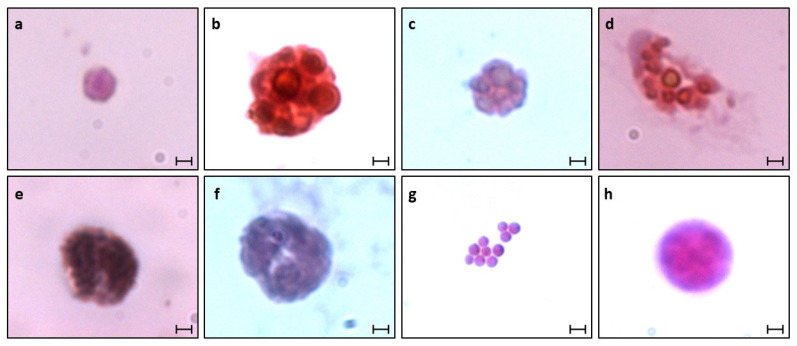
Haemocytes of *B. schlosseri* and thraustochytrid cells fixed and then stained with H&E: (**a**) hemoblast cell; (**b**) macrophage-like cell; (**c**) morula cell; (**d**) granular amebocyte cell; (**e**) pigment cell; (**f**) nephrocyte cell; (**g**,**h**) thraustochytrid cells, (**g**) mononucleated cell cluster composed of ten individual cells and (**h**) sporangium cell. Scale bars: 2 µm in (**a**–**f**,**h**), 5 µm in (**g**).

**Figure 5 cells-12-01709-f005:**
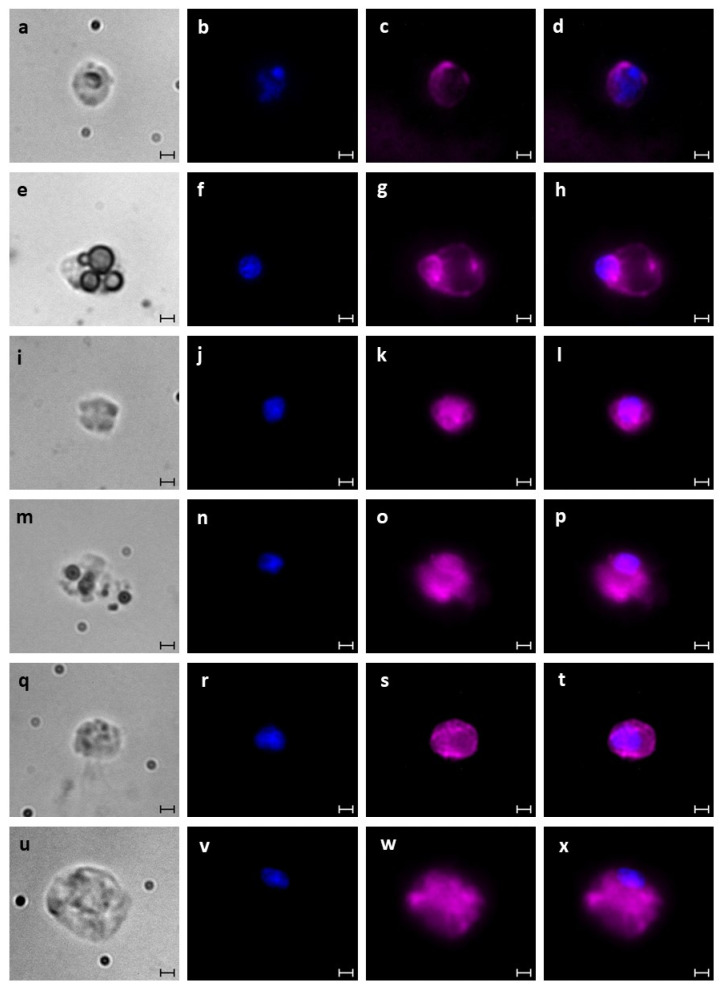
Haemocytes of *B. schlosseri* stained with Hoechst and DiD upon fixation. (**a**–**d**) Haemoblast cell imaged with bright field channel (**a**) stained with Hoechst (**b**) and DiD (**c**); (**d**) merged image. (**e**–**h**) Macrophage-like cell imaged with bright field channel (**e**) stained with Hoechst (**f**) and DiD (**g**); (**h**) merged image. (**i**–**l**) Morula cell imaged with a bright field channel (**i**) stained with Hoechst (**j**) and DiD (**k**); (**l**) merged image. (**m**–**p**) Granular amoebocyte cell imaged with a bright field channel (**m**) stained with Hoechst (**n**) and DiD (**o**); (**p**) merged image. (**q**–**t**) Pigment cell imaged with a bright field channel (**q**) stained with Hoechst (**r**) and DiD (**s**); (**t**) merged image. (**u**–**x**) Nephrocyte cell imaged with a bright field channel (**u**) stained with Hoechst (**v**) and DiD (**w**); (**x**) merged image. Scale bars = 2 µm.

**Figure 6 cells-12-01709-f006:**
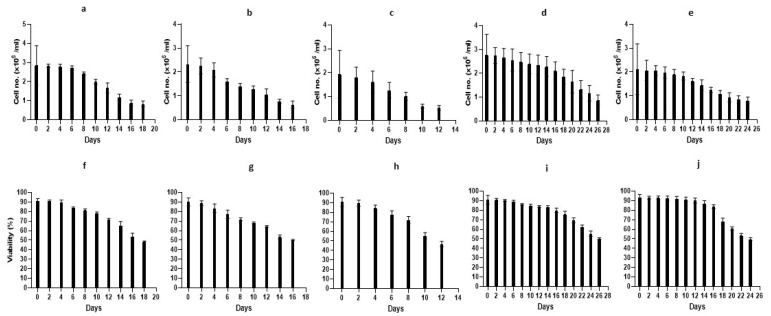
Cell numbers (10^6^ cells/mL) and cell viability (%) of primary cultures in the five media versions, up to 26 days from onset. Cell numbers in medium TGM1 (**a**), TGM2 (**b**), TGM3 (**c**), TGM4 (**d**), and TGM5 (**e**). Cell viabilities in medium TGM1 (**f**), TGM2 (**g**), TGM3 (**h**), TGM4 (**i**), and TGM5 (**j**).

**Figure 7 cells-12-01709-f007:**
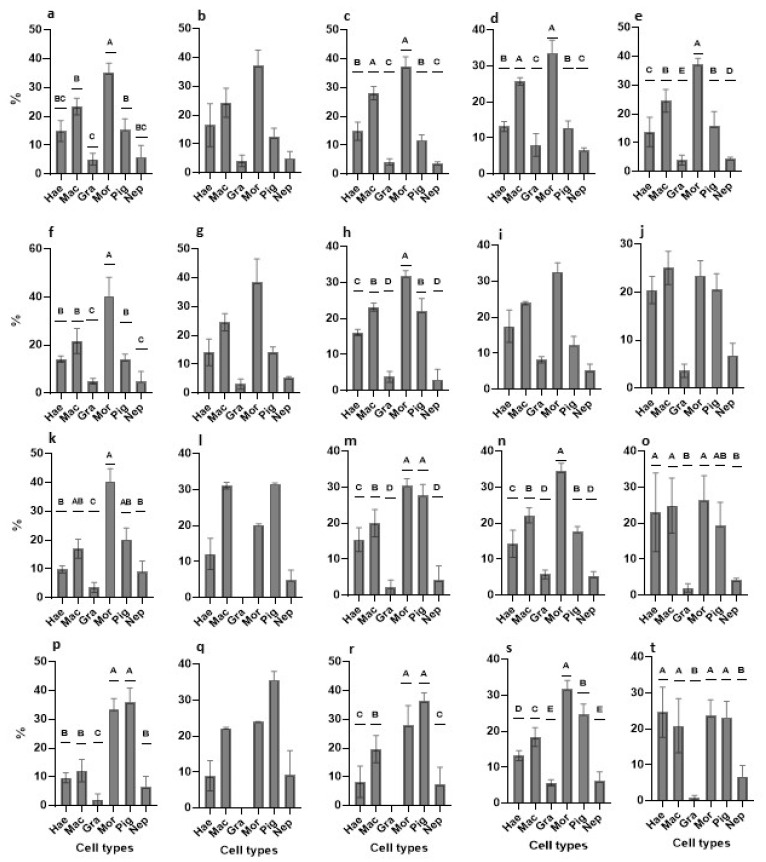
Distributions (%) of cell types in *B. schlosseri* primary cultures under five media types at onset, 24 h, 3 days, and 8 days from initiation. (**a**–**e**) At onset in medium TGM1 (**a**), TGM2 (**b**), TGM3 (**c**), TGM4 (**d**), and TGM5 (**e**). (**f**–**j**) At 24 h in medium TGM1 (**f**), TGM2 (**g**), TGM3 (**h**), TGM4 (**i**), and TGM5 (**j**). (**k–o**) at day 3 in medium TGM1 (**k**), TGM2 (**l**), TGM3 (**m**), TGM4 (**n**), and TGM5 (**o**). (**p**–**t**) At day 8 in medium TGM1 (**p**), TGM2 (**q**), TGM3 (**r**), TGM4 (**s**), and TGM5 (**t**). Hae, haemoblasts; Mac, macrophage-like cells; Gra, granular amoebocytes; Mor, morula cells; Pig, pigment cells; Nep, nephrocytes. Capital letters above bars represent statistically different groups (*p* < 0.001, *p* < 0.05; one-way ANOVA using post hoc comparison Bonferroni and Tukey HSD).

**Figure 8 cells-12-01709-f008:**
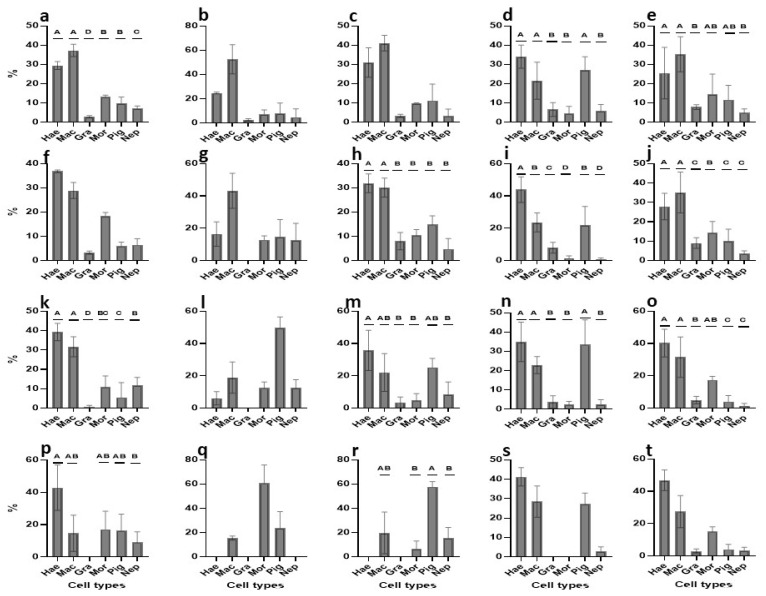
Distributions (%) of PCNA^+^ cells in *B. schlosseri* primary cultures under five media conditions at onset, 24 h, 3 days, and 8 days from initiation. (**a**–**e**) At onset in medium TGM1 (**a**), TGM2 (**b**), TGM3 (**c**), TGM4 (**d**), and TGM5 (**e**). (**f**–**j**) At 24 h in medium TGM1 (**f**), TGM2 (**g**), TGM3 (**h**)**,** TGM4 (**i**), and TGM5 (**j**). (**k**–**o**) At day 3 in medium TGM1 (**k**), TGM2 (**l**), TGM3 (**m**), TGM4 (**n**), and TGM5 (**o**). (**p**–**t**) At day 8 in medium TGM1 (**p**), TGM2 (**q**), TGM3 (**r**), TGM4 (**s**), and TGM5 (**t**). Hae, haemoblasts; Mac, macrophage-like cells; Gra, granular amoebocytes; Mor, morula cells; Pig, pigment cells; Nep, nephrocytes. Capital letters above bars refer to statistically different groups (*p* < 0.001, *p* < 0.05; one-way ANOVA using post hoc comparison Bonferroni and Tukey HSD).

**Table 1 cells-12-01709-t001:** Yields of *B. schlosseri* blood cell extractions and cell viability in the five media formulations. WS = washing solution.

Medium	Exp. No.	No. of Colonies	Blastogenic Stage	No. of Zooids	WS Type	Cell Numbers/Exp n ± STD (×10^6^)	Viability ± STD (%)
TGM1	1	2	A, D	7, 11	WS1	3.1 ± 1.1	92.1 ± 1.3
	2	3	B, C, D	12, 8, 8		2.9 ± 1.4	92.3 ± 2.7
	3	4	A, C, C, D	15, 10, 8, 12		2.7 ± 0.7	90.4 ± 1.7
TGM2	1	3	B, C, D	10, 11, 9	WS1	2.6 ± 0.6	88.6 ± 2.9
	2	3	C, C, D	8, 8, 12		2.1 ± 0.9	91.6 ± 4.9
TGM3	1	2	A, D	14, 11	WS2	1.3 ± 0.33	94.6 ± 2.0
	2	2	B, D	8, 10		2.4 ± 0.9	87.1 ± 6.4
	3	3	B, C, D	7, 8, 12		2.1 ± 1.2	90.8 ± 2.9
TGM4	1	2	A, C	11, 12	WS3	2.8 ± 0.9	91.1 ± 1.4
	2	2	B, C	10, 7		2.3 ± 0.5	92.8 ± 1.3
	3	2	C, D	15, 10		3.2 ± 0.9	90.7 ± 6.7
	4	3	C, C, D	11, 8, 12		2.8 ± 1.0	89.9 ± 6.2
TGM5	1	3	C, C, D	7, 8, 10	WS1	1.6 ± 0.6	94 ± 3.4
	2	3	B, C, D	10, 12, 15		2.6 ± 1.4	91.9 ± 4.0
	3	3	A, D, D	11, 8, 7		2.2 ± 1.0	94.3 ± 1.6

**Table 2 cells-12-01709-t002:** Major findings for primary cultures (cell type abundance and PCNA^+^ cells) across the five media versions, ranging from the onset to 8-day-old cultures. Hae, haemoblasts; mac, macrophage-like cells; mor, morula cells; pig, pigment cells.

Medium	Major Outcomes	Onset	24-h	Day-3	Day-8
TGM1	abundant cell types	mac, mor	mac, mor	mor	mor, pig
	PCNA^+^ cells	hae, mac	hae, mac, mor	hae, mac	hae
TGM2	abundant cell types	mac, mor	mac, mor	mac, mor, pig	mac, mor, pig
	PCNA^+^ cells	hae, mac	mac	pig	mor
TGM3	abundant cell types	mac, mor	mac, mor, pig	hae, mac, mor, pig	mac, mor, pig
	PCNA^+^ cells	hae, mac	hae, mac	hae, mac, pig	pig
TGM4	abundant cell types	mac, mor	hae, mac, mor	mac, mor, pig	hae, mac, mor, pig
	PCNA^+^ cells	hae, mac, pig	mac, mor, pig	hae, mac, pig	hae, mac, pig
TGM5	abundant cell types	mac, mor	hae, mac, mor, pig	hae, mac, mor, pig	hae, mac, mor, pig
	PCNA^+^ cells	hae, mac	hae, mac	hae, mac, mor	hae, mac, mor

## Data Availability

All data are presented in Appendix A.

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
