# Peer review of "Improved Media Formulations for Primary Cell Cultures Derived from a Colonial Urochordate"

_cells, 2023, doi:10.3390/cells12131709_

Round 1

Reviewer 1 Report

This manuscript will serve as an invaluable reference to anyone seeking to study not only urochordate cell biology but the cell biology and cell culture of any marine invertebrate.  It sets the standard for how experiments (and, in particular, marine invertebrate nutrient media selection experiments) should be designed.   The methods are adequately detailed, and the results are explained clearly and supported extremely well with the images, graphs, and tables.   

The last two sentences of the manuscript--lines 587-591--should be emphasized a bit more.   The authors understate their discovery and classification of cell types using natural fluorescence, as well as how significant their findings are in terms of developing cell cultures from different cell lines of marine invertebrates.   While a "given" in mammalian and insect cell culture, this is novel in marine invertebrate cell culture.

The English language is excellent.  There are only minor edits--an occasional "an" or "the" should be inserted in some of the sentences.   And I found just a couple of errors:

line 335--thraustochytrid is misspelled

line 518--should read "need to develop"

Figures 6, 7, & 8  appear (to me) to be a bit fuzzy--can they be sharpened?  If it's possible to add color to the bars in Figures 7 and 8?   

Reviewer 2 Report

General consideration.

The manuscript “Improved media formulations for primary cell cultures derived from a colonial urochordate” is a very interesting and important work both in the field of comparative immunology, (for a basic and comparative research approach) and for biotechnological applications.

The paper is well structured and written. However, I have some considerations and curiosity. The authors in their approach use the PCNA immunostaining for investigating cell proliferation, why other proliferation assays, as BrdU assay or better the EdU assay, have not been used? These might have been more informative allowing to identify both the cells that underwent proliferation and those that are proliferating.

An additional consideration is that other parameters important to take into account, in setting in vitro culture of blood cells, is to verify also if their functions are conserved. For example, it would have been useful to test the phagocytic activity of macrophages and granular amoebocytes for determining if the culture conditions do not affect the function of these cells. This could be an important point to clarify in the discussion, including also what still needs to be improved and demonstrated for obtaining a blood cell culture ready to be used for in vitro assays and for biotechnological approaches.

Other minor comments:

Material and Methods:

-     Section 2.3 Blood cell observation: the authors clearly explain how they checked for thraustochytrids and yeast/fungi contaminations, but they do not mention bacteria contamination that they should check with 16S PCR or media’s spot on bacterial culture plates. Also, please report how often the contamination checks have been performed.

-     Line 124: Table S1 is not formatted properly in the supplementary file. Also, for making clearer and immediate the formulation of the media, it would be better to also add the percentage of each component in each cell culture medium.

-     Section 2.9 Statistics: Please describe better on what have been done the statistical analysis. The One-way ANOVA have been performed on the results of proliferation and viability assays? On the different time point of cell culture for each medium formulation? Make a clear description of the conditions tested in One-way ANOVA analysis.

Results:

Section 3.1 Identification of B. schlosseri blood cell populations:

-     Lines 232-234: please explain better why these staining have been used for the characterization of the cell types.

-     Figure 3, pictures of granular amoebocytes are missed.

-     Figure 4 needs a better magnification of the cell type otherwise it is difficult to appreciate the description made in the result section (lines 234-253).

-     Figure 5i needs a better focus of the cell type.

Section 3.2 Cultivation of B. schlosseri blood cells

-     Section 3.3 and Table 1 should go after figure 5 that belong to the previous section (3.1).

-     Line 282: specify the criterion used for the choice of the WS for a culture medium. I guess that it depends on the formulation of antibiotic used, that should be the same between the two solutions.

-     Figure 6: the graphics are difficult to read, please enlarge them.

-     Figure 6, 7 and 8: it would be easier and immediate to read the figures if you indicate on the graphics also the corresponding cell culture medium and the time points tested instead to report them just in the legends.

Also, Figures 7 and 8 are grainy and sometimes the letter that labels the graphic overlaps with the capital letters that indicate the statistics results.

-     Line 286: why do you perform counting of cell types and proliferative assay only at four time points, with the last one at day 8, considering that the cells are kept in culture for 26 days?

-     Line 287: please make clearer how it is performed the PCNA analysis, I guess that you count, for each cell type, the total number and then those that are PCNA positive. Clarify this somewhere in the manuscript, or in this section or in the Material and Methods section.

-     Line 314: Figure S5f, Calcofluor white staining should be fluorescent, it is strange that this picture is in bright field, please clarify or include a better picture.

-     Lines 315-319: clarify how the three and four distinct groups have been determined, it is not clear. This is not clear neither in the figure legends (Figure 7 and 8) where you mention two p-value indicating statistical different groups but then there are 3-4 groups indicated with capital letters. This consideration is extended to all the “distinct groups” you have identified in this work.

-     Figure S6 seems cut off.

-     Line 158: change “to developed” with “to develop”.

Discussion

-       Line 526: which conditions of culture have been used initially that were unsuccessful? It could be important to discuss this point, make comparison between old settings and this more successful blood cell culture setting.

-       Line 531: Would be better to specify again what is the basic medium used.

-       Paragraph 549-564: Another important points to discuss with a clearer statement are i) which is/are the medium/media that is/are better for the B. schlosseri blood cell culture and why, ii) make hypothesis on which components of the media are important for the viability and proliferation and which is the best antibiotic condition to use or the better pH of the medium.

Reviewer 3 Report

Cultivating cells from marine invertebrates is very challenging, and until recently, no single continuous cell line has been established for any aquatic invertebrates (except a marine sponge cell line). In this manuscript, the authors assessed how various Botryllus blood cell types respond to in-vitro conditions by utilizing five different cell culture media. The findings are in line with recent advances in marine invertebrate cell cultures. This is an interesting report, and it’s a piece of necessary work for the field.

Below are some concerns:

1.     In lines 194 and 195, the authors used anti-rabbit PCNA to measure the B. schlosseri PCNA. Do the PCNA sequences between these two species share high similarities?

2.     Table 1, what’s the rationale for making these combinations? For instance, TGM1 was only tested with WS1, but not for WS2 and WS3.

3.     From Figure 6, it’s clear that the total number of cells was not increased under five media types. The authors also measured the distribution of cell types and claimed that there was a medium-dependent increase in the proliferation of distinct blood cell types. However, the proportion increase of some cell types may be due to the death of other cell types.
